# DeBERTa: Decoding-enhanced BERT with Disentangled Attention

**Pengcheng He[1], Xiaodong Liu[2], Jianfeng Gao[2], Weizhu Chen[1]**
[1] Microsoft Dynamics 365 AI     [2] Microsoft Research
{penhe,xiaodl,jfgao,wzchen}@microsoft.com

## Abstract

Recent progress in pre-trained neural language models has significantly improved the performance of many natural language processing (NLP) tasks. In this paper we propose a new model architecture **DeBERTa** (**D**ecoding-**e**nhanced **BERT** with disentangled **a**ttention) that improves the BERT and RoBERTa models using two novel techniques. The first is the disentangled attention mechanism, where each word is represented using two vectors that encode its content and position, respectively, and the attention weights among words are computed using disentangled matrices on their contents and relative positions, respectively. Second, an enhanced mask decoder is used to incorporate absolute positions in the decoding layer to predict the masked tokens in model pre-training. In addition, a new virtual adversarial training method is used for fine-tuning to improve models' generalization. We show that these techniques significantly improve the efficiency of model pre-training and the performance of both natural language understand (NLU) and natural langauge generation (NLG) downstream tasks. Compared to RoBERTa-Large, a DeBERTa model trained on half of the training data performs consistently better on a wide range of NLP tasks, achieving improvements on MNLI by +0.9% (90.2% vs. 91.1%), on SQuAD v2.0 by +2.3% (88.4% vs. 90.7%) and RACE by +3.6% (83.2% vs. 86.8%). Notably, we scale up DeBERTa by training a larger version that consists of 48 Transform layers with 1.5 billion parameters. The significant performance boost makes the single DeBERTa model surpass the human performance on the SuperGLUE benchmark (Wang et al., 2019a) for the first time in terms of macro-average score (89.9 versus 89.8), and the ensemble DeBERTa model sits atop the SuperGLUE leaderboard as of January 6, 2021, outperforming the human baseline by a decent margin (90.3 versus 89.8). The pre-trained DeBERTa models and the source code were released at: https://github.com/microsoft/DeBERTa[1].

## 1 Introduction

The Transformer has become the most effective neural network architecture for neural language modeling. Unlike recurrent neural networks (RNNs) that process text in sequence, Transformers apply self-attention to compute in parallel every word from the input text an attention weight that gauges the influence each word has on another, thus allowing for much more parallelization than RNNs for large-scale model training (Vaswani et al., 2017). Since 2018, we have seen the rise of a set of large-scale Transformer-based Pre-trained Language Models (PLMs), such as GPT (Radford et al., 2019; Brown et al., 2020), BERT (Devlin et al., 2019), RoBERTa (Liu et al., 2019c), XLNet (Yang et al., 2019), UniLM (Dong et al., 2019), ELECTRA (Clark et al., 2020), T5 (Raffel et al., 2020), ALUM (Liu et al., 2020), StructBERT (Wang et al., 2019c) and ERINE (Sun et al., 2019) . These PLMs have been fine-tuned using task-specific labels and created new state of the art in many downstream natural language processing (NLP) tasks (Liu et al., 2019b; Minaee et al., 2020; Jiang et al., 2020; He et al., 2019a;b; Shen et al., 2020).

---

[1]Our code and models are also available at HuggingFace Transformers: https://github.com/huggingface/transformers, https://huggingface.co/models?filter=deberta

In this paper, we propose a new Transformer-based neural language model **DeBERTa** (**D**ecoding-**e**nhanced **BERT** with disentangled **a**ttention), which improves previous state-of-the-art PLMs using two novel techniques: a disentangled attention mechanism, and an enhanced mask decoder.

**Disentangled attention.**   Unlike BERT where each word in the input layer is represented using a vector which is the sum of its word (content) embedding and position embedding, each word in DeBERTa is represented using two vectors that encode its content and position, respectively, and the attention weights among words are computed using disentangled matrices based on their contents and relative positions, respectively. This is motivated by the observation that the attention weight of a word pair depends on not only their contents but their relative positions. For example, the dependency between the words "deep" and "learning" is much stronger when they occur next to each other than when they occur in different sentences.

**Enhanced mask decoder.**   Like BERT, DeBERTa is pre-trained using masked language modeling (MLM). MLM is a fill-in-the-blank task, where a model is taught to use the words surrounding a mask token to predict what the masked word should be. DeBERTa uses the content and position information of the context words for MLM. The disentangled attention mechanism already considers the contents and relative positions of the context words, but not the absolute positions of these words, which in many cases are crucial for the prediction. Consider the sentence "a new store opened beside the new mall" with the italicized words "store" and "mall" masked for prediction. Although the local contexts of the two words are similar, they play different syntactic roles in the sentence. (Here, the subject of the sentence is "store" not "mall," for example.) These syntactical nuances depend, to a large degree, upon the words' absolute positions in the sentence, and so it is important to account for a word's absolute position in the language modeling process. DeBERTa incorporates absolute word position embeddings right before the softmax layer where the model decodes the masked words based on the aggregated contextual embeddings of word contents and positions.

In addition, we propose a new virtual adversarial training method for fine-tuning PLMs to downstream NLP tasks. The method is effective in improving models' generalization.

We show through a comprehensive empirical study that these techniques substantially improve the efficiency of pre-training and the performance of downstream tasks. In the NLU tasks, compared to RoBERTa-Large, a DeBERTa model trained on half the training data performs consistently better on a wide range of NLP tasks, achieving improvements on MNLI by +0.9% (90.2% vs. 91.1%), on SQuAD v2.0 by +2.3%(88.4% vs. 90.7%), and RACE by +3.6% (83.2% vs. 86.8%). In the NLG tasks, DeBERTa reduces the perplexity from 21.6 to 19.5 on the Wikitext-103 dataset. We further scale up DeBERTa by pre-training a larger model that consists of 48 Transformer layers with 1.5 billion parameters. The single 1.5B-parameter DeBERTa model substantially outperforms T5 with 11 billion parameters on the SuperGLUE benchmark (Wang et al., 2019a) by 0.6%(89.3% vs. 89.9%), and surpasses the human baseline (89.9 vs. 89.8) for the first time. The ensemble DeBERTa model sits atop the SuperGLUE leaderboard as of January 6, 2021, outperforming the human baseline by a decent margin (90.3 versus 89.8).

## 2 BACKGROUND

### 2.1 TRANSFORMER

A Transformer-based language model is composed of stacked Transformer blocks (Vaswani et al., 2017). Each block contains a multi-head self-attention layer followed by a fully connected positional feed-forward network. The standard self-attention mechanism lacks a natural way to encode word position information. Thus, existing approaches add a positional bias to each input word embedding so that each input word is represented by a vector whose value depends on its content and position. The positional bias can be implemented using absolute position embedding (Vaswani et al., 2017; Radford et al., 2019; Devlin et al., 2019) or relative position embedding (Huang et al., 2018; Yang et al., 2019). It has been shown that relative position representations are more effective for natural language understanding and generation tasks (Dai et al., 2019; Shaw et al., 2018). The proposed disentangled attention mechanism differs from all existing approaches in that we represent each input word using two separate vectors that encode a word's content and position, respectively, and

attention weights among words are computed using disentangled matrices on their contents and relative positions, respectively.

## 2.2 Masked Language Model

Large-scale Transformer-based PLMs are typically pre-trained on large amounts of text to learn contextual word representations using a self-supervision objective, known as Masked Language Model (MLM) (Devlin et al., 2019). Specifically, given a sequence $\boldsymbol{X} = \{x_i\}$, we corrupt it into $\tilde{\boldsymbol{X}}$ by masking 15% of its tokens at random and then train a language model parameterized by $\theta$ to reconstruct $\boldsymbol{X}$ by predicting the masked tokens $\tilde{x}$ conditioned on $\tilde{\boldsymbol{X}}$:

$$\max_\theta \log p_\theta(\boldsymbol{X}|\tilde{\boldsymbol{X}}) = \max_\theta \sum_{i \in \mathcal{C}} \log p_\theta(\tilde{x}_i = x_i|\tilde{\boldsymbol{X}}) \tag{1}$$

where $\mathcal{C}$ is the index set of the masked tokens in the sequence. The authors of BERT propose to keep 10% of the masked tokens unchanged, another 10% replaced with randomly picked tokens and the rest replaced with the `[MASK]` token.

## 3 The DeBERTa Architecture

### 3.1 Disentangled Attention: A Two-Vector Approach to Content and Position Embedding

For a token at position $i$ in a sequence, we represent it using two vectors, $\{\boldsymbol{H_i}\}$ and $\{\boldsymbol{P_{i|j}}\}$, which represent its content and relative position with the token at position $j$, respectively. The calculation of the cross attention score between tokens $i$ and $j$ can be decomposed into four components as

$$\begin{aligned} A_{i,j} &= \{\boldsymbol{H_i}, \boldsymbol{P_{i|j}}\} \times \{\boldsymbol{H_j}, \boldsymbol{P_{j|i}}\}^\intercal \\ &= \boldsymbol{H_i}\boldsymbol{H_j^\intercal} + \boldsymbol{H_i}\boldsymbol{P_{j|i}^\intercal} + \boldsymbol{P_{i|j}}\boldsymbol{H_j^\intercal} + \boldsymbol{P_{i|j}}\boldsymbol{P_{j|i}^\intercal} \end{aligned} \tag{2}$$

That is, the attention weight of a word pair can be computed as a sum of four attention scores using disentangled matrices on their contents and positions as *content-to-content*, *content-to-position*, *position-to-content*, and *position-to-position* [2].

Existing approaches to relative position encoding use a separate embedding matrix to compute the relative position bias in computing attention weights (Shaw et al., 2018; Huang et al., 2018). This is equivalent to computing the attention weights using only the content-to-content and content-to-position terms in equation 2. We argue that the position-to-content term is also important since the attention weight of a word pair depends not only on their contents but on their relative positions, which can only be fully modeled using both the content-to-position and position-to-content terms. Since we use *relative* position embedding, the position-to-position term does not provide much additional information and is removed from equation 2 in our implementation.

Taking single-head attention as an example, the standard self-attention operation (Vaswani et al., 2017) can be formulated as:

$$\boldsymbol{Q} = \boldsymbol{H}\boldsymbol{W_q}, \boldsymbol{K} = \boldsymbol{H}\boldsymbol{W_k}, \boldsymbol{V} = \boldsymbol{H}\boldsymbol{W_v}, \boldsymbol{A} = \frac{\boldsymbol{Q}\boldsymbol{K}^\intercal}{\sqrt{d}}$$

$$\boldsymbol{H_o} = \text{softmax}(\boldsymbol{A})\boldsymbol{V}$$

where $\boldsymbol{H} \in R^{N \times d}$ represents the input hidden vectors, $\boldsymbol{H_o} \in R^{N \times d}$ the output of self-attention, $\boldsymbol{W_q}, \boldsymbol{W_k}, \boldsymbol{W_v} \in R^{d \times d}$ the projection matrices, $\boldsymbol{A} \in R^{N \times N}$ the attention matrix, $N$ the length of the input sequence, and $d$ the dimension of hidden states.

Denote $k$ as the maximum relative distance, $\delta(i, j) \in [0, 2k)$ as the relative distance from token $i$ to token $j$, which is defined as:

$$\delta(i, j) = \begin{cases} 0 & \text{for} & i - j \leqslant -k \\ 2k - 1 & \text{for} & i - j \geqslant k \\ i - j + k & \text{others.} \end{cases} \tag{3}$$

---

[2]In this sense, our model shares some similarity to Tensor Product Representation (Smolensky, 1990; Schlag et al., 2019; Chen et al., 2019) where a word is represented using a tensor product of its filler (content) vector and its role (position) vector.

We can represent the disentangled self-attention with relative position bias as equation 4, where $\boldsymbol{Q_c}, \boldsymbol{K_c}$ and $\boldsymbol{V_c}$ are the projected content vectors generated using projection matrices $\boldsymbol{W_{q,c}}, \boldsymbol{W_{k,c}}, \boldsymbol{W_{v,c}} \in R^{d \times d}$ respectively, $\boldsymbol{P} \in R^{2k \times d}$ represents the relative position embedding vectors shared across all layers (i.e., staying fixed during forward propagation), and $\boldsymbol{Q_r}$ and $\boldsymbol{K_r}$ are projected relative position vectors generated using projection matrices $\boldsymbol{W_{q,r}}, \boldsymbol{W_{k,r}} \in R^{d \times d}$, respectively.

$$\boldsymbol{Q_c} = \boldsymbol{H}\boldsymbol{W_{q,c}}, \boldsymbol{K_c} = \boldsymbol{H}\boldsymbol{W_{k,c}}, \boldsymbol{V_c} = \boldsymbol{H}\boldsymbol{W_{v,c}}, \boldsymbol{Q_r} = \boldsymbol{P}\boldsymbol{W_{q,r}}, \boldsymbol{K_r} = \boldsymbol{P}\boldsymbol{W_{k,r}}$$

$$\tilde{A}_{i,j} = \underbrace{\boldsymbol{Q}_i^c \boldsymbol{K}_j^{c\mathsf{T}}}_{\text{(a) content-to-content}} + \underbrace{\boldsymbol{Q}_i^c \boldsymbol{K}_{\delta(i,j)}^{r}{}^{\mathsf{T}}}_{\text{(b) content-to-position}} + \underbrace{\boldsymbol{K}_j^c \boldsymbol{Q}_{\delta(j,i)}^{r}{}^{\mathsf{T}}}_{\text{(c) position-to-content}} \tag{4}$$

$$\boldsymbol{H_o} = \text{softmax}(\frac{\tilde{\boldsymbol{A}}}{\sqrt{3d}})\boldsymbol{V_c}$$

$\tilde{A}_{i,j}$ is the element of attention matrix $\tilde{\boldsymbol{A}}$, representing the attention score from token $i$ to token $j$. $\boldsymbol{Q}_i^c$ is the $i$-th row of $\boldsymbol{Q_c}$. $\boldsymbol{K}_j^c$ is the $j$-th row of $\boldsymbol{K_c}$. $\boldsymbol{K}_{\delta(i,j)}^r$ is the $\delta(i,j)$-th row of $\boldsymbol{K_r}$ with regarding to relative distance $\delta(i,j)$. $\boldsymbol{Q}_{\delta(j,i)}^r$ is the $\delta(j,i)$-th row of $\boldsymbol{Q_r}$ with regarding to relative distance $\delta(j,i)$. Note that we use $\delta(j,i)$ rather than $\delta(i,j)$ here. This is because for a given position $i$, position-to-content computes the attention weight of the key content at $j$ with respect to the query position at $i$, thus the relative distance is $\delta(j,i)$. The position-to-content term is calculated as $\boldsymbol{K}_j^c \boldsymbol{Q}_{\delta(j,i)}^{r}{}^{\mathsf{T}}$. The content-to-position term is calculated in a similar way.

Finally, we apply a scaling factor of $\frac{1}{\sqrt{3d}}$ on $\tilde{\boldsymbol{A}}$. The factor is important for stabilizing model training (Vaswani et al., 2017), especially for large-scale PLMs.

---

**Algorithm 1** Disentangled Attention

---

**Input:** Hidden state $\boldsymbol{H}$, relative distance embedding $\boldsymbol{P}$, relative distance matrix $\boldsymbol{\delta}$. Content projection matrix $\boldsymbol{W_{k,c}}, \boldsymbol{W_{q,c}}, \boldsymbol{W_{v,c}}$, position projection matrix $\boldsymbol{W_{k,r}}, \boldsymbol{W_{q,r}}$.
1: $\boldsymbol{K_c} = \boldsymbol{H}\boldsymbol{W_{k,c}}, \boldsymbol{Q_c} = \boldsymbol{H}\boldsymbol{W_{q,c}}, \boldsymbol{V_c} = \boldsymbol{H}\boldsymbol{W_{v,c}}, \boldsymbol{K_r} = \boldsymbol{P}\boldsymbol{W_{k,r}}, \boldsymbol{Q_r} = \boldsymbol{P}\boldsymbol{W_{q,r}}$
2: $\boldsymbol{A_{c \to c}} = \boldsymbol{Q_c}\boldsymbol{K_c^{\mathsf{T}}}$
3: **for** $i = 0, ..., N-1$ **do**
4: $\quad$ $\tilde{\boldsymbol{A}}_{c \to p}[i, :] = \boldsymbol{Q_c}[i, :]\boldsymbol{K_r^{\mathsf{T}}}$
5: **end for**
6: **for** $i = 0, ..., N-1$ **do**
7: $\quad$ **for** $j = 0, ..., N-1$ **do**
8: $\quad\quad$ $\boldsymbol{A_{c \to p}}[i, j] = \tilde{\boldsymbol{A}}_{c \to p}[i, \boldsymbol{\delta}[i, j]]$
9: $\quad$ **end for**
10: **end for**
11: **for** $j = 0, ..., N-1$ **do**
12: $\quad$ $\tilde{\boldsymbol{A}}_{p \to c}[:, j] = \boldsymbol{K_c}[j, :]\boldsymbol{Q_r^{\mathsf{T}}}$
13: **end for**
14: **for** $j = 0, ..., N-1$ **do**
15: $\quad$ **for** $i = 0, ..., N-1$ **do**
16: $\quad\quad$ $\boldsymbol{A_{p \to c}}[i, j] = \tilde{\boldsymbol{A}}_{p \to c}[\boldsymbol{\delta}[j, i], j]$
17: $\quad$ **end for**
18: **end for**
19: $\tilde{\boldsymbol{A}} = \boldsymbol{A_{c \to c}} + \boldsymbol{A_{c \to p}} + \boldsymbol{A_{p \to c}}$
20: $\boldsymbol{H_o} = \text{softmax}(\frac{\tilde{\boldsymbol{A}}}{\sqrt{3d}})\boldsymbol{V_c}$
**Output:** $\boldsymbol{H_o}$

---

### 3.1.1 EFFICIENT IMPLEMENTATION

For an input sequence of length $N$, it requires a space complexity of $O(N^2 d)$ (Shaw et al., 2018; Huang et al., 2018; Dai et al., 2019) to store the relative position embedding for each token. However, taking content-to-position as an example, we note that since $\delta(i,j) \in [0, 2k)$ and the embeddings

of all possible relative positions are always a subset of $\boldsymbol{K_r} \in R^{2k \times d}$, then we can reuse $\boldsymbol{K_r}$ in the attention calculation for all the queries.

In our experiments, we set the maximum relative distance $k$ to 512 for pre-training. The disentangled attention weights can be computed efficiently using Algorithm 1. Let $\boldsymbol{\delta}$ be the relative position matrix according to equation 3, i.e., $\boldsymbol{\delta}[i,j] = \delta(i,j)$. Instead of allocating a different relative position embedding matrix for each query, we multiply each $query$ vector $\boldsymbol{Q_c}[i,:]$ by $\boldsymbol{K_r^\intercal} \in R^{d \times 2k}$, as in line $3-5$. Then, we extract the attention weight using the relative position matrix $\boldsymbol{\delta}$ as the index, as in line $6-10$. To compute the position-to-content attention score, we calculate $\tilde{\boldsymbol{A}}_{\boldsymbol{p \rightarrow c}}[:,j]$, i.e., the column vector of the attention matrix $\tilde{\boldsymbol{A}}_{\boldsymbol{p \rightarrow c}}$, by multiplying each $key$ vector $\boldsymbol{K_c}[j,:]$ by $\boldsymbol{Q_r^\intercal}$, as in line $11-13$. Finally, we extract the corresponding attention score via the relative position matrix $\boldsymbol{\delta}$ as the index, as in line $14-18$. In this way, we do not need to allocate memory to store a relative position embedding for each query and thus reduce the space complexity to $O(kd)$ (for storing $\boldsymbol{K_r}$ and $\boldsymbol{Q_r}$).

## 3.2 Enhanced Mask Decoder Accounts for Absolute Word Positions

DeBERTa is pretrained using MLM, where a model is trained to use the words surrounding a mask token to predict what the masked word should be. DeBERTa uses the content and position information of the context words for MLM. The disentangled attention mechanism already considers the contents and relative positions of the context words, but not the absolute positions of these words, which in many cases are crucial for the prediction.

Given a sentence "a new **store** opened beside the new **mall**" with the words "store" and "mall" masked for prediction. Using only the local context (e.g., relative positions and surrounding words) is insufficient for the model to distinguish *store* and *mall* in this sentence, since both follow the word *new* with the same relative positions. To address this limitation, the model needs to take into account absolute positions, as complement information to the relative positions. For example, the subject of the sentence is "store" not "mall". These syntactical nuances depend, to a large degree, upon the words' absolute positions in the sentence.

There are two methods of incorporating absolute positions. The BERT model incorporates absolute positions in the input layer. In DeBERTa, we incorporate them right after all the Transformer layers but before the *softmax* layer for masked token prediction, as shown in Figure 2. In this way, DeBERTa captures the relative positions in all the Transformer layers and only uses absolute positions as complementary information when decoding the masked words. Thus, we call DeBERTa's decoding component an Enhanced Mask Decoder (EMD). In the empirical study, we compare these two methods of incorporating absolute positions and observe that EMD works much better. We conjecture that the early incorporation of absolute positions used by BERT might undesirably hamper the model from learning sufficient information of relative positions. In addition, EMD also enables us to introduce other useful information, in addition to positions, for pre-training. We leave it to future work.

## 4 Scale Invariant Fine-Tuning

This section presents a new virtual adversarial training algorithm, Scale-invariant-Fine-Tuning (SiFT), a variant to the algorithm described in Miyato et al. (2018); Jiang et al. (2020), for fine-tuning.

Virtual adversarial training is a regularization method for improving models' generalization. It does so by improving a model's robustness to adversarial examples, which are created by making small perturbations to the input. The model is regularized so that when given a task-specific example, the model produces the same output distribution as it produces on an adversarial perturbation of that example.

For NLP tasks, the perturbation is applied to the word embedding instead of the original word sequence. However, the value ranges (norms) of the embedding vectors vary among different words and models. The variance gets larger for bigger models with billions of parameters, leading to some instability of adversarial training.

Inspired by layer normalization (Ba et al., 2016), we propose the SiFT algorithm that improves the training stability by applying the perturbations to the *normalized* word embeddings. Specifically, when fine-tuning DeBERTa to a downstream NLP task in our experiments, SiFT first normalizes the word embedding vectors into stochastic vectors, and then applies the perturbation to the normalized embedding vectors. We find that the normalization substantially improves the performance of the fine-tuned models. The improvement is more prominent for larger DeBERTa models. Note that we **only** apply SiFT to DeBERTa$_{1.5B}$ on SuperGLUE tasks in our experiments and we will provide a more comprehensive study of SiFT in our future work.

## 5 EXPERIMENT

This section reports DeBERTa results on various NLU tasks.

### 5.1 MAIN RESULTS ON NLU TASKS

Following previous studies of PLMs, we report results using large and base models.

#### 5.1.1 PERFORMANCE ON LARGE MODELS

| Model | CoLA Mcc | QQP Acc | MNLI-m/mm Acc | SST-2 Acc | STS-B Corr | QNLI Acc | RTE Acc | MRPC Acc | Avg. |
|---|---|---|---|---|---|---|---|---|---|
| BERT$_{large}$ | 60.6 | 91.3 | 86.6/- | 93.2 | 90.0 | 92.3 | 70.4 | 88.0 | 84.05 |
| RoBERTa$_{large}$ | 68.0 | 92.2 | 90.2/90.2 | 96.4 | 92.4 | 93.9 | 86.6 | 90.9 | 88.82 |
| XLNet$_{large}$ | 69.0 | 92.3 | 90.8/90.8 | 97.0 | 92.5 | 94.9 | 85.9 | 90.8 | 89.15 |
| ELECTRA$_{large}$ | 69.1 | 92.4 | 90.9/- | 96.9 | 92.6 | 95.0 | 88.0 | 90.8 | 89.46 |
| DeBERTa$_{large}$ | 70.5 | 92.3 | 91.1/91.1 | 96.8 | 92.8 | 95.3 | 88.3 | 91.9 | 90.00 |

Table 1: Comparison results on the GLUE development set.

We pre-train our large models following the setting of BERT (Devlin et al., 2019), except that we use the BPE vocabulary of Radford et al. (2019); Liu et al. (2019c). For training data, we use Wikipedia (English Wikipedia dump[3]; 12GB), BookCorpus (Zhu et al., 2015) (6GB), OPENWEBTEXT (public Reddit content (Gokaslan & Cohen, 2019); 38GB), and STORIES (a subset of CommonCrawl (Trinh & Le, 2018); 31GB). The total data size after data deduplication (Shoeybi et al., 2019) is about 78G. Refer to Appendix A.2 for a detailed description of the pre-training dataset.

We use 6 DGX-2 machines (96 V100 GPUs) to train the models. A single model trained with 2K batch size and 1M steps takes about 20 days. Refer to Appendix A for the detailed hyperparamters.

We summarize the results on eight NLU tasks of GLUE (Wang et al., 2019b) in Table 1, where DeBERTa is compared DeBERTa with previous Transform-based PLMs of similar structures (i.e. 24 layers with hidden size of 1024) including BERT, RoBERTa, XLNet, ALBERT and ELECTRA. Note that RoBERTa, XLNet and ELECTRA are pre-trained on 160G training data while DeBERTa is pre-trained on 78G training data. RoBERTa and XLNet are pre-trained for 500K steps with 8K samples in a step, which amounts to four billion training samples. DeBERTa is pre-trained for one million steps with 2K samples in each step. This amounts to two billion training samples, approximately half of either RoBERTa or XLNet. Table 1 shows that compared to BERT and RoBERTa, DeBERTa performs consistently better across all the tasks. Meanwhile, DeBERTa outperforms XLNet in six out of eight tasks. Particularly, the improvements on MRPC (1.1% over XLNet and 1.0% over RoBERTa), RTE (2.4% over XLNet and 1.7% over RoBERTa) and CoLA (1.5% over XLNet and 2.5% over RoBERTa) are significant. DeBERTa also outperforms other SOTA PLMs, i.e., ELECTRA$_{large}$ and XLNet$_{large}$, in terms of average GLUE score.

Among all GLUE tasks, MNLI is most often used as an indicative task to monitor the research progress of PLMs. DeBERTa significantly outperforms all existing PLMs of similar size on MNLI and creates a new state of the art.

---

[3]https://dumps.wikimedia.org/enwiki/

| Model | MNLI-m/mm Acc | SQuAD v1.1 F1/EM | SQuAD v2.0 F1/EM | RACE Acc | ReCoRD F1/EM | SWAG Acc | NER F1 |
|---|---|---|---|---|---|---|---|
| BERT$_{large}$ | 86.6/- | 90.9/84.1 | 81.8/79.0 | 72.0 | - | 86.6 | 92.8 |
| ALBERT$_{large}$ | 86.5/- | 91.8/85.2 | 84.9/81.8 | 75.2 | - | - | - |
| RoBERTa$_{large}$ | 90.2/90.2 | 94.6/88.9 | 89.4/86.5 | 83.2 | 90.6/90.0 | 89.9 | 93.4 |
| XLNet$_{large}$ | 90.8/90.8 | 95.1/89.7 | 90.6/87.9 | 85.4 | - | - | - |
| Megatron$_{336M}$ | 89.7/90.0 | 94.2/88.0 | 88.1/84.8 | 83.0 | - | - | - |
| DeBERTa$_{large}$ | **91.1/91.1** | **95.5/90.1** | **90.7/88.0** | **86.8** | **91.4/91.0** | 90.8 | **93.8** |
| ALBERT$_{xxlarge}$ | 90.8/- | 94.8/89.3 | 90.2/87.4 | 86.5 | - | - | - |
| Megatron$_{1.3B}$ | 90.9/91.0 | 94.9/89.1 | 90.2/87.1 | 87.3 | - | - | - |
| Megatron$_{3.9B}$ | 91.4/91.4 | 95.5/90.0 | 91.2/88.5 | 89.5 | - | - | - |

Table 2: Results on MNLI in/out-domain, SQuAD v1.1, SQuAD v2.0, RACE, ReCoRD, SWAG, CoNLL 2003 NER development set. Note that missing results in literature are signified by "-".

In addition to GLUE, DeBERTa is evaluated on three categories of NLU benchmarks: (1) Question Answering: SQuAD v1.1 (Rajpurkar et al., 2016), SQuAD v2.0 (Rajpurkar et al., 2018), RACE (Lai et al., 2017), ReCoRD (Zhang et al., 2018) and SWAG (Zellers et al., 2018); (2) Natural Language Inference: MNLI (Williams et al., 2018); and (3) NER: CoNLL-2003. For comparison, we include ALBERT$_{xxlarge}$ (Lan et al., 2019) [4] and Megatron (Shoeybi et al., 2019) with three different model sizes, denoted as Megatron$_{336M}$, Megatron$_{1.3B}$ and Megatron$_{3.9B}$, respectively, which are trained using the same dataset as RoBERTa. Note that Megatron$_{336M}$ has a similar model size as other models mentioned above[5].

We summarize the results in Table 2. Compared to the previous SOTA PLMs with a similar model size (i.e., BERT, RoBERTa, XLNet, ALBERT$_{large}$, and Megatron$_{336M}$), DeBERTa shows superior performance in all seven tasks. Taking the RACE benchmark as an example, DeBERTa significantly outperforms XLNet by +1.4% (86.8% vs. 85.4%). Although Megatron$_{1.3B}$ is three times larger than DeBERTa, DeBERTa outperforms it in three of the four benchmarks. We further report DeBERTa on text generation tasks in Appendix A.4.

### 5.1.2 Performance on Base Models

Our setting for base model pre-training is similar to that for large models. The base model structure follows that of the BERT base model, i.e., $L = 12$, $H = 768$, $A = 12$. We use 4 DGX-2 with 64 V100 GPUs to train the base model. It takes 10 days to finish a single pre-training of 1M training steps with batch size 2048. We train DeBERTa using the same 78G dataset, and compare it to RoBERTa and XLNet trained on 160G text data.

We summarize the base model results in Table 3. Across all three tasks, DeBERTa consistently outperforms RoBERTa and XLNet by a larger margin than that in large models. For example, on MNLI-m, DeBERTa$_{base}$ obtains +1.2% (88.8% vs. 87.6%) over RoBERTa$_{base}$, and +2% (88.8% vs. 86.8%) over XLNet$_{base}$.

| Model | MNLI-m/mm (Acc) | SQuAD v1.1 (F1/EM) | SQuAD v2.0 (F1/EM) |
|---|---|---|---|
| RoBERTa$_{base}$ | 87.6/- | 91.5/84.6 | 83.7/80.5 |
| XLNet$_{base}$ | 86.8/- | -/- | -/80.2 |
| DeBERTa$_{base}$ | **88.8/88.5** | **93.1/87.2** | **86.2/83.1** |

Table 3: Results on MNLI in/out-domain (m/mm), SQuAD v1.1 and v2.0 development set.

---

[4]The hidden dimension of ALBERT$_{xxlarge}$ is 4 times of DeBERTa and the computation cost is about 4 times of DeBERTa.

[5]T5 (Raffel et al., 2020) has more parameters (11B). Raffel et al. (2020) only report the test results of T5 which are not comparable with other models.

## 5.2 MODEL ANALYSIS

In this section, we first present an ablation study to quantify the relative contributions of different components introduced in DeBERTa. Then, we study the convergence property to characterize the model training efficiency. We run experiments for analysis using the base model setting: a model is pre-trained using the Wikipedia + Bookcorpus dataset for 1M steps with batch size 256 in 7 days on a DGX-2 machine with 16 V-100 GPUs. Due to space limit, we visualize the different attention patterns of DeBERTa and RoBERTa in Appendix A.7.

### 5.2.1 ABLATION STUDY

To verify our experimental setting, we pre-train the RoBERTa base model from scratch. The re-pre-trained RoBERTa model is denoted as RoBERTa-ReImp$_{base}$. To investigate the relative contributions of different components in DeBERTa, we develop three variations:

- -EMD is the DeBERTa base model without EMD.
- -C2P is the DeBERTa base model without the content-to-position term ((c) in Eq. 4).
- -P2C is the DeBERTa base model without the position-to-content term ((b) in Eq. 4). As XLNet also uses the relative position bias, this model is close to XLNet plus EMD.

| **Model** | MNLI-m/mm Acc | SQuAD v1.1 F1/EM | SQuAD v2.0 F1/EM | RACE Acc |
|---|---|---|---|---|
| BERT$_{base}$ Devlin et al. (2019) | 84.3/84.7 | 88.5/81.0 | 76.3/73.7 | 65.0 |
| RoBERTa$_{base}$ Liu et al. (2019c) | 84.7/- | 90.6/- | 79.7/- | 65.6 |
| XLNet$_{base}$ Yang et al. (2019) | 85.8/85.4 | -/- | 81.3/78.5 | 66.7 |
| RoBERTa-ReImp$_{base}$ | 84.9/85.1 | 91.1/84.8 | 79.5/76.0 | 66.8 |
| DeBERTa$_{base}$ | **86.3/86.2** | **92.1/86.1** | **82.5/79.3** | **71.7** |
| -EMD | 86.1/86.1 | 91.8/85.8 | 81.3/78.0 | 70.3 |
| -C2P | 85.9/85.7 | 91.6/85.8 | 81.3/78.3 | 69.3 |
| -P2C | 86.0/85.8 | 91.7/85.7 | 80.8/77.6 | 69.6 |
| -(EMD+C2P) | 85.8/85.9 | 91.5/85.3 | 80.3/77.2 | 68.1 |
| -(EMD+P2C) | 85.8/85.8 | 91.3/85.1 | 80.2/77.1 | 68.5 |

Table 4: Ablation study of the DeBERTa base model.

Table 4 summarizes the results on four benchmark datasets. First, RoBERTa-ReImp performs similarly to RoBERTa across all benchmark datasets, verfiying that our setting is reasonable. Second, we see that removing any one component in DeBERTa results in a sheer performance drop. For instance, removing EMD (-EMD) results in a loss of 1.4% (71.7% vs. 70.3%) on RACE, 0.3% (92.1% vs. 91.8%) on SQuAD v1.1, 1.2% (82.5% vs. 81.3%) on SQuAD v2.0, 0.2% (86.3% vs. 86.1%) and 0.1% (86.2% vs. 86.1%) on MNLI-m/mm, respectively. Similarly, removing either *content-to-position* or *position-to-content* leads to inferior performance in all the benchmarks. As expected, removing two components results in even more substantial loss in performance.

## 5.3 SCALE UP TO 1.5 BILLION PARAMETERS

Larger pre-trained models have shown better generalization results (Raffel et al., 2020; Brown et al., 2020; Shoeybi et al., 2019). Thus, we have built a larger version of DeBERTa with 1.5 billion parameters, denoted as DeBERTa$_{1.5B}$. The model consists of 48 layers with a hidden size of 1,536 and 24 attention heads [6]. DeBERTa$_{1.5B}$ is trained on a pre-training dataset amounting to 160G, similar to that in Liu et al. (2019c), with a new vocabulary of size 128K constructed using the dataset.

To train DeBERTa$_{1.5B}$, we optimize the model architecture as follows. First, we share the projection matrices of relative position embedding $W_{k,r}, W_{q,r}$ with $W_{k,c}, W_{q,c}$, respectively, in all attention layers to reduce the number of model parameters. Our ablation study in Table 13 on base models shows that the projection matrix sharing reduces the model size while retaining the model performance.

---

[6]See Table 8 in Appendix for the model hyperparameters.

Second, a convolution layer is added aside the first Transformer layer to induce n-gram knowledge of sub-word encodings and their outputs are summed up before feeding to the next Transformer layer [7].

Table 5 reports the test results of SuperGLUE (Wang et al., 2019a) which is one of the most popular NLU benchmarks. SuperGLUE consists of a wide of NLU tasks, including Question Answering (Clark et al., 2019; Khashabi et al., 2018; Zhang et al., 2018), Natural Language Inference (Dagan et al., 2006; Bar-Haim et al., 2006; Giampiccolo et al., 2007; Bentivogli et al., 2009), Word Sense Disambiguation (Pilehvar & Camacho-Collados, 2019), and Reasoning (Levesque et al., 2011; Roemmele et al., 2011). Since its release in 2019, top research teams around the world have been developing large-scale PLMs that have driven striking performance improvement on SuperGLUE.

The significant performance boost due to scaling DeBERTa to a larger model makes the single DeBERTa$_{1.5B}$ surpass the human performance on SuperGLUE for the first time in terms of macro-average score (89.9 versus 89.8) as of December 29, 2020, and the ensemble DeBERTa model (DeBERTa$_{Ensemble}$) sits atop the SuperGLUE benchmark rankings as of January 6, 2021, outperforming the human baseline by a decent margin (90.3 versus 89.8). Compared to T5, which consists of 11 billion parameters, the 1.5-billion-parameter DeBERTa is much more energy efficient to train and maintain, and it is easier to compress and deploy to apps of various settings.

| Model | BoolQ Acc | CB F1/Acc | COPA Acc | MultiRC F1a/EM | ReCoRD F1/EM | RTE Acc | WiC Acc | WSC Acc | Average Score |
|---|---|---|---|---|---|---|---|---|---|
| RoBERTa$_{large}$ | 87.1 | 90.5/95.2 | 90.6 | 84.4/52.5 | 90.6/90.0 | 88.2 | 69.9 | 89.0 | 84.6 |
| NEXHA-Plus | 87.8 | 94.4/96.0 | 93.6 | 84.6/55.1 | 90.1/89.6 | 89.1 | 74.6 | 93.2 | 86.7 |
| T5$_{11B}$ | 91.2 | 93.9/96.8 | 94.8 | 88.1/63.3 | 94.1/93.4 | 92.5 | 76.9 | 93.8 | 89.3 |
| T5$_{11B}$+Meena | **91.3** | **95.8/97.6** | 97.4 | 88.3/63.0 | 94.2/93.5 | 92.7 | **77.9** | 95.9 | 90.2 |
| Human | 89.0 | 95.8/98.9 | 100.0 | 81.8/51.9 | 91.7/91.3 | 93.6 | 80.0 | 100.0 | 89.8 |
| DeBERTa$_{1.5B}$+SiFT | 90.4 | 94.9/97.2 | 96.8 | **88.2/63.7** | **94.5/94.1** | **93.2** | 76.4 | **95.9** | 89.9 |
| DeBERTa$_{Ensemble}$ | 90.4 | 95.7/97.6 | **98.4** | 88.2/63.7 | 94.5/94.1 | 93.2 | 77.5 | 95.9 | **90.3** |

Table 5: SuperGLUE test set results scored using the SuperGLUE evaluation server. All the results are obtained from https://super.gluebenchmark.com on January 6, 2021.

## 6 CONCLUSIONS

This paper presents a new model architecture DeBERTa (Decoding-enhanced BERT with disentangled attention) that improves the BERT and RoBERTa models using two novel techniques. The first is the disentangled attention mechanism, where each word is represented using two vectors that encode its content and position, respectively, and the attention weights among words are computed using disentangled matrices on their contents and relative positions, respectively. The second is an enhanced mask decoder which incorporates absolute positions in the decoding layer to predict the masked tokens in model pre-training. In addition, a new virtual adversarial training method is used for fine-tuning to improve model's generalization on downstream tasks.

We show through a comprehensive empirical study that these techniques significantly improve the efficiency of model pre-training and the performance of downstream tasks. The DeBERTa model with 1.5 billion parameters surpasses the human performance on the SuperGLUE benchmark for the first time in terms of macro-average score.

DeBERTa surpassing human performance on SuperGLUE marks an important milestone toward general AI. Despite its promising results on SuperGLUE, the model is by no means reaching the human-level intelligence of NLU. Humans are extremely good at leveraging the knowledge learned from different tasks to solve a new task with no or little task-specific demonstration. This is referred to as *compositional generalization*, the ability to generalize to novel compositions (new tasks) of familiar constituents (subtasks or basic problem-solving skills). Moving forward, it is worth exploring how to make DeBERTa incorporate compositional structures in a more explicit manner, which could allow combining neural and symbolic computation of natural language similar to what humans do.

---

[7]Please refer to Table 12 in Appendix A.6 for the ablation study of different model sizes, and Table 13 in Appendix A.6 for the ablation study of new modifications.

## 7 ACKNOWLEDGMENTS

We thank Jade Huang and Nikos Karampatziakis for proofreading the paper and providing insightful comments. We thank Yoyo Liang, Saksham Singhal, Xia Song, and Saurabh Tiwary for their help with large-scale model training. We also thank the anonymous reviewers for valuable discussions.

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

## A    APPENDIX

### A.1    DATASET

| Corpus | Task | #Train | #Dev | #Test | #Label | Metrics |
|--------|------|--------|------|-------|--------|---------|
| **General Language Understanding Evaluation (GLUE)** | | | | | | |
| CoLA | Acceptability | 8.5k | 1k | 1k | 2 | Matthews corr |
| SST | Sentiment | 67k | 872 | 1.8k | 2 | Accuracy |
| MNLI | NLI | 393k | 20k | 20k | 3 | Accuracy |
| RTE | NLI | 2.5k | 276 | 3k | 2 | Accuracy |
| WNLI | NLI | 634 | 71 | 146 | 2 | Accuracy |
| QQP | Paraphrase | 364k | 40k | 391k | 2 | Accuracy/F1 |
| MRPC | Paraphrase | 3.7k | 408 | 1.7k | 2 | Accuracy/F1 |
| QNLI | QA/NLI | 108k | 5.7k | 5.7k | 2 | Accuracy |
| STS-B | Similarity | 7k | 1.5k | 1.4k | 1 | Pearson/Spearman corr |
| **SuperGLUE** | | | | | | |
| WSC | Coreference | 554k | 104 | 146 | 2 | Accuracy |
| BoolQ | QA | 9,427 | 3,270 | 3,245 | 2 | Accuracy |
| COPA | QA | 400k | 100 | 500 | 2 | Accuracy |
| CB | NLI | 250 | 57 | 250 | 3 | Accuracy/F1 |
| RTE | NLI | 2.5k | 276 | 3k | 2 | Accuracy |
| WiC | WSD | 2.5k | 276 | 3k | 2 | Accuracy |
| ReCoRD | MRC | 101k | 10k | 10k | - | Exact Match (EM)/F1 |
| MultiRC | Multiple choice | 5,100 | 953 | 1,800 | - | Exact Match (EM)/F1 |
| **Question Answering** | | | | | | |
| SQuAD v1.1 | MRC | 87.6k | 10.5k | 9.5k | - | Exact Match (EM)/F1 |
| SQuAD v2.0 | MRC | 130.3k | 11.9k | 8.9k | - | Exact Match (EM)/F1 |
| RACE | MRC | 87,866 | 4,887 | 4,934 | 4 | Accuracy |
| SWAG | Multiple choice | 73.5k | 20k | 20k | 4 | Accuracy |
| **Token Classification** | | | | | | |
| CoNLL 2003 | NER | 14,987 | 3,466 | 3,684 | 8 | F1 |

Table 6: Summary information of the NLP application benchmarks.

• **GLUE**. The General Language Understanding Evaluation (GLUE) benchmark is a collection of nine natural language understanding (NLU) tasks. As shown in Table 6, it includes question answering (Rajpurkar et al., 2016), linguistic acceptability (Warstadt et al., 2018), sentiment analysis (Socher et al., 2013), text similarity (Cer et al., 2017), paraphrase detection (Dolan & Brockett, 2005), and natural language inference (NLI) (Dagan et al., 2006; Bar-Haim et al., 2006; Giampiccolo et al., 2007; Bentivogli et al., 2009; Levesque et al., 2012; Williams et al., 2018). The diversity of the tasks makes GLUE very suitable for evaluating the generalization and robustness of NLU models.

• **SuperGLUE**. SuperGLUE is an extension of the GLUE benchmark, but more difficult, which is a collection of eight NLU tasks. It covers a various of tasks including question answering (Zhang et al., 2018; Clark et al., 2019; Khashabi et al., 2018), natural language inference (Dagan et al., 2006; Bar-Haim et al., 2006; Giampiccolo et al., 2007; Bentivogli et al., 2009; De Marneffe et al., 2019), coreference resolution (Levesque et al., 2012) and word sense disambiguation (Pilehvar & Camacho-Collados, 2019).

• **RACE** is a large-scale machine reading comprehension dataset, collected from English examinations in China, which are designed for middle school and high school students (Lai et al., 2017).

• **SQuAD v1.1/v2.0** is the Stanford Question Answering Dataset (SQuAD) v1.1 and v2.0 (Rajpurkar et al., 2016; 2018) are popular machine reading comprehension benchmarks. Their passages come from approximately 500 Wikipedia articles and the questions and answers are obtained by crowd-sourcing. The SQuAD v2.0 dataset includes unanswerable questions about the same paragraphs.

• **SWAG** is a large-scale adversarial dataset for the task of grounded commonsense inference, which unifies natural language inference and physically grounded reasoning (Zellers et al., 2018). SWAG consists of 113k multiple choice questions about grounded situations.

• **CoNLL 2003** is an English dataset consisting of text from a wide variety of sources. It has 4 types of named entity.

## A.2 PRE-TRAINING DATASET

For DeBERTa pre-training, we use Wikipedia (English Wikipedia dump[8]; 12GB), BookCorpus (Zhu et al., 2015) [9] (6GB), OPENWEBTEXT (public Reddit content (Gokaslan & Cohen, 2019); 38GB) and STORIES[10] (a subset of CommonCrawl (Trinh & Le, 2018); 31GB). The total data size after data deduplication(Shoeybi et al., 2019) is about 78GB. For pre-training, we also sample 5% training data as the validation set to monitor the training process. Table 7 compares datasets used in different pre-trained models.

| Model | Wiki+Book 16GB | OpenWebText 38GB | Stories 31GB | CC-News 76GB | Giga5 16GB | ClueWeb 19GB | Common Crawl 110GB |
|---|---|---|---|---|---|---|---|
| BERT | ✓ | | | | | | |
| XLNet | ✓ | | | | ✓ | ✓ | ✓ |
| RoBERTa | ✓ | ✓ | ✓ | ✓ | | | |
| DeBERTa | ✓ | ✓ | ✓ | | | | |
| DeBERTa$_{1.5B}$ | ✓ | ✓ | ✓ | ✓ | | | |

Table 7: Comparison of the pre-training data.

## A.3 IMPLEMENTATION DETAILS

Following RoBERTa (Liu et al., 2019c), we adopt dynamic data batching. We also include span masking (Joshi et al., 2020) as an additional masking strategy with the span size up to three. We list the detailed hyperparameters of pre-training in Table 8. For pre-training, we use Adam (Kingma & Ba, 2014) as the optimizer with weight decay (Loshchilov & Hutter, 2018). For fine-tuning, even though we can get better and robust results with RAdam(Liu et al., 2019a) on some tasks, e.g. CoLA, RTE and RACE, we use Adam(Kingma & Ba, 2014) as the optimizer for a fair comparison. For fine-tuning, we train each task with a hyper-parameter search procedure, each run takes about 1-2 hours on a DGX-2 node. All the hyper-parameters are presented in Table 9. The model selection is based on the performance on the task-specific development sets.

Our code is implemented based on Huggingface Transformers[11], FairSeq[12] and Megatron (Shoeybi et al., 2019)[13].

### A.3.1 PRE-TRAINING EFFICIENCY

To investigate the efficiency of model pre-training, we plot the performance of the fine-tuned model on downstream tasks as a function of the number of pre-training steps. As shown in Figure 1, for RoBERTa-ReImp$_{base}$ and DeBERTa$_{base}$, we dump a checkpoint every 150K pre-training steps, and then fine-tune the checkpoint on two representative downstream tasks, MNLI and SQuAD v2.0, and then report the accuracy and F1 score, respectively. As a reference, we also report the final model performance of both the original RoBERTa$_{base}$ (Liu et al., 2019c) and XLNet$_{base}$ (Yang et al., 2019). The results show that DeBERTa$_{base}$ consistently outperforms RoBERTa-ReImp$_{base}$ during the course of pre-training.

---

[8]https://dumps.wikimedia.org/enwiki/

[9]https://github.com/butsugiri/homemade_bookcorpus

[10]https://github.com/tensorflow/models/tree/master/research/lm_commonsense

[11]https://github.com/huggingface/transformers

[12]https://github.com/pytorch/fairseq

[13]https://github.com/NVIDIA/Megatron-LM

| Hyper-parameter | DeBERTa$_{1.5B}$ | DeBERTa$_{large}$ | DeBERTa$_{base}$ | DeBERTa$_{base-ablation}$ |
|---|---|---|---|---|
| Number of Layers | 48 | 24 | 12 | 12 |
| Hidden size | 1536 | 1024 | 768 | 768 |
| FNN inner hidden size | 6144 | 4096 | 3072 | 3072 |
| Attention Heads | 24 | 16 | 12 | 12 |
| Attention Head size | 64 | 64 | 64 | 64 |
| Dropout | 0.1 | 0.1 | 0.1 | 0.1 |
| Warmup Steps | 10k | 10k | 10k | 10k |
| Learning Rates | 1.5e-4 | 2e-4 | 2e-4 | 1e-4 |
| Batch Size | 2k | 2k | 2k | 256 |
| Weight Decay | 0.01 | 0.01 | 0.01 | 0.01 |
| Max Steps | 1M | 1M | 1M | 1M |
| Learning Rate Decay | Linear | Linear | Linear | Linear |
| Adam $\epsilon$ | 1e-6 | 1e-6 | 1e-6 | 1e-6 |
| Adam $\beta_1$ | 0.9 | 0.9 | 0.9 | 0.9 |
| Adam $\beta_2$ | 0.999 | 0.999 | 0.999 | 0.999 |
| Gradient Clipping | 1.0 | 1.0 | 1.0 | 1.0 |
| Number of DGX-2 nodes | 16 | 6 | 4 | 1 |
| Training Time | 30 days | 20 days | 10 days | 7 days |

Table 8: Hyper-parameters for pre-training DeBERTa.

| Hyper-parameter | DeBERTa$_{1.5B}$ | DeBERTa$_{large}$ | DeBERTa$_{base}$ |
|---|---|---|---|
| Dropout of task layer | {0,0.15,0.3} | {0,0.1,0.15} | {0,0.1,0.15} |
| Warmup Steps | {50,100,500,1000} | {50,100,500,1000} | {50,100,500,1000} |
| Learning Rates | {1e-6, 3e-6, 5e-6} | {5e-6, 8e-6, 9e-6, 1e-5} | {1.5e-5,2e-5, 3e-5, 4e-5} |
| Batch Size | {16,32,64} | {16,32,48,64} | {16,32,48,64} |
| Weight Decay | 0.01 | 0.01 | |
| Maximun Training Epochs | 10 | 10 | 10 |
| Learning Rate Decay | Linear | Linear | Linear |
| Adam $\epsilon$ | 1e-6 | 1e-6 | 1e-6 |
| Adam $\beta_1$ | 0.9 | 0.9 | 0.9 |
| Adam $\beta_2$ | 0.999 | 0.999 | 0.999 |
| Gradient Clipping | 1.0 | 1.0 | 1.0 |

Table 9: Hyper-parameters for fine-tuning DeBERTa on down-streaming tasks.

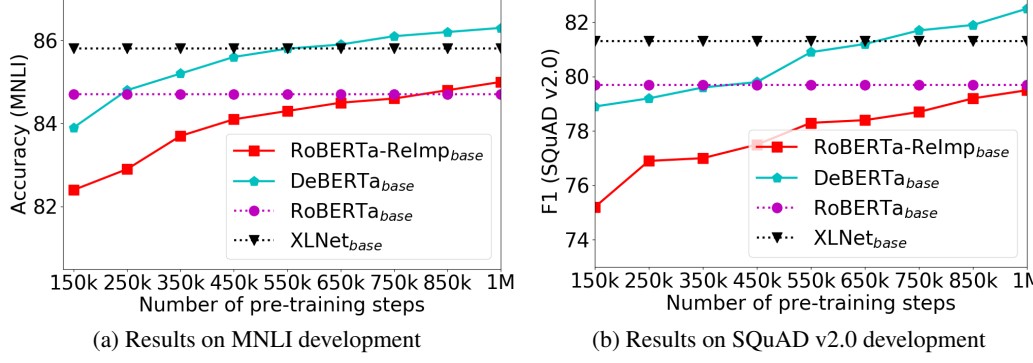

(a) Results on MNLI development

(b) Results on SQuAD v2.0 development

Figure 1: Pre-training performance curve between DeBERTa and its counterparts on the MNLI and SQuAD v2.0 development set.

## A.4 Main Results on Generation Tasks

In addition to NLU tasks, DeBERTa can also be extended to handle NLG tasks. To allow DeBERTa operating like an auto-regressive model for text generation, we use a triangular matrix for self-attention and set the upper triangular part of the self-attention mask to $-\infty$, following Dong et al. (2019).

We evaluate DeBERTa on the task of auto-regressive language model (ARLM) using Wikitext-103 (Merity et al., 2016). To do so, we train a new version of DeBERTa, denoted as DeBERTa-MT. It is jointly pre-trained using the MLM and ARLM tasks as in UniLM (Dong et al., 2019). The pre-training hyper-parameters follows that of DeBERTa$_{base}$ except that we use fewer training steps (200k). For comparison, we use RoBERTa as baseline, and include GPT-2 and Transformer-XL as additional references. DeBERTa-AP is a variant of DeBERTa where absolute position embeddings are incorporated in the input layer as RoBERTa. For a fair comparison, all these models are base models pre-trained in a similar setting.

| Model | RoBERTa | DeBERTa-AP | DeBERTa | DeBERTa-MT | GPT-2 | Transformer-XL |
|-------|---------|------------|---------|------------|-------|----------------|
| Dev PPL | 21.6 | 20.7 | 20.5 | **19.5** | - | 23.1 |
| Test PPL | 21.6 | 20.0 | 19.9 | **19.5** | 37.50 | 24 |

Table 10: Language model results in perplexity (lower is better) on Wikitext-103 .

Table 10 summarizes the results on Wikitext-103. We see that DeBERTa$_{base}$ obtains lower perplexities on both dev and test data, and joint training using MLM and ARLM reduces perplexity further. That DeBERTa-AP is inferior to DeBERTa indicates that it is more effective to incorporate absolute position embeddings of words in the decoding layer as the EMD in DeBERTa than in the input layer as RoBERTa.

## A.5 Handling long sequence input

With relative position bias, we choose to truncate the maximum relative distance to $k$ as in equation 3. Thus in each layer, each token can attend directly to at most $2(k-1)$ tokens and itself. By stacking Transformer layers, each token in the $l-$th layer can attend to at most $(2k-1)l$ tokens implicitly. Taking DeBERTa$_{large}$ as an example, where $k = 512, L = 24$, in theory, the maximum sequence length that can be handled is 24,528. This is a byproduct benefit of our design choice and we find it beneficial for the RACE task. A comparison of long sequence effect on the RACE task is shown in Table 11.

| Sequence length | Middle | High | Accuracy |
|-----------------|--------|------|----------|
| 512 | 88.8 | 85.0 | 86.3 |
| 768 | 88.7 | 86.3 | 86.8 |

Table 11: The effect of handling long sequence input for RACE task with DeBERTa

Long sequence handling is an active research area. There have been a lot of studies where the Transformer architecture is extended for long sequence handling(Beltagy et al., 2020; Kitaev et al., 2019; Child et al., 2019; Dai et al., 2019). One of our future research directions is to extend DeBERTa to deal with extremely long sequences.

## A.6 Performance improvements of different model scales

In this subsection, we study the effect of different model sizes applied to large models on GLUE. Table 12 summarizes the results, showing that larger models can obtain a better result and SiFT also improves the model performance consistently.

| Model | CoLA Mcc | QQP Acc | MNLI-m/mm Acc | SST-2 Acc | STS-B Corr | QNLI Acc | RTE Acc | MRPC Acc | Avg. |
|---|---|---|---|---|---|---|---|---|---|
| DeBERTa$_{large}$ | 70.5 | 92.3 | 91.1/91.1 | 96.8 | 92.8 | 95.3 | 88.3 | 91.9 | 90.00 |
| DeBERTa$_{900M}$ | 71.1 | 92.3 | 91.7/91.6 | **97.5** | 92.0 | 95.8 | 93.5 | 93.1 | 90.86 |
| DeBERTa$_{1.5B}$ | 72.0 | 92.7 | 91.7/91.9 | 97.2 | 92.9 | 96.0 | 93.9 | 92.0 | 91.17 |
| DeBERTa$_{1.5B}$+SiFT | **73.5** | **93.0** | **92.0/92.1** | 97.5 | **93.2** | **96.5** | **96.5** | **93.2** | **91.93** |

Table 12: Comparison results of DeBERTa models with different sizes on the GLUE development set.

| Model | Parameters | MNLI-m/mm Acc | SQuAD v1.1 F1/EM | SQuAD v2.0 F1/EM |
|---|---|---|---|---|
| RoBERTa-ReImp$_{base}$ | 120M | 84.9/85.1 | 91.1/84.8 | 79.5/76.0 |
| DeBERTa$_{base}$ | 134M | 86.3/86.2 | 92.1/86.1 | 82.5/79.3 |
| + ShareProjection | 120M | 86.3/86.3 | 92.2/86.2 | 82.3/79.5 |
| + Conv | 122M | 86.3/86.5 | 92.5/86.4 | 82.5/79.7 |
| + 128k Vocab | 190M | 86.7/86.9 | 93.1/86.8 | 83.0/80.1 |

Table 13: Ablation study of the additional modifications in DeBERTa$_{1.5B}$ and DeBERTa$_{900M}$ models. Note that we progressively add each component on the top of DeBERTa$_{base}$.

## A.7 MODEL COMPLEXITY

With the disentangled attention mechanism, we introduce three additional sets of parameters $W_{q,r}, W_{k,r} \in R^{d \times d}$ and $P \in R^{2k \times d}$. The total increase in model parameters is $2L \times d^2 + 2k \times d$. For the large model ($d = 1024, L = 24, k = 512$), this amounts to about $49M$ additional parameters, an increase of $13\%$. For the base model($d = 768, L = 12, k = 512$), this amounts to $14M$ additional parameters, an increase of $12\%$. However, by sharing the projection matrix between content and position embedding, i.e. $W_{q,r} = W_{q,c}, W_{k,r} = W_{k,c}$, the number of parameters of DeBERTa is the same as RoBERTa. Our experiment on base model shows that the results are almost the same, as in Table 13.

The additional computational complexity is $O(Nkd)$ due to the calculation of the additional *position-to-content* and *content-to-position* attention scores. Compared with BERT or RoBERTa, this increases the computational cost by $30\%$. Compared with XLNet which also uses relative position embedding, the increase of computational cost is about $15\%$. A further optimization by fusing the attention computation kernel can significantly reduce this additional cost. For $EMD$, since the decoder in pre-training only reconstructs the masked tokens, it does not introduce additional computational cost for unmasked tokens. In the situation where $15\%$ tokens are masked and we use only two decoder layers, the additional cost is $0.15 \times 2/L$ which results in an additional computational cost of only $3\%$ for base model($L = 12$) and $2\%$ for large model($L = 24$) in EMD.

## A.8 ADDITIONAL DETAILS OF ENHANCED MASK DECODER

The structure of EMD is shown in Figure 2b. There are two inputs for EMD, (i.e., $I, H$). $H$ denotes the hidden states from the previous Transformer layer, and $I$ can be any necessary information for decoding, e.g., $H$, absolute position embedding or output from previous EMD layer. $n$ denotes $n$ stacked layers of EMD where the output of each EMD layer will be the input $I$ for next EMD layer and the output of last EMD layer will be fed to the language model head directly. The $n$ layers can share the same weight. In our experiment we share the same weight for $n = 2$ layers to reduce the number of parameters and use absolute position embedding as $I$ of the first EMD layer. When $I = H$ and $n = 1$, EMD is the same as the BERT decoder layer. However, EMD is more general and flexible as it can take various types of input information for decoding.

## A.9 ATTENTION PATTERNS

To visualize how DeBERTa operates differently from RoBERTa, we present in Figure 3 the attention patterns (taken in the last self-attention layers) of RoBERTa, DeBERTa and three DeBERTa variants.

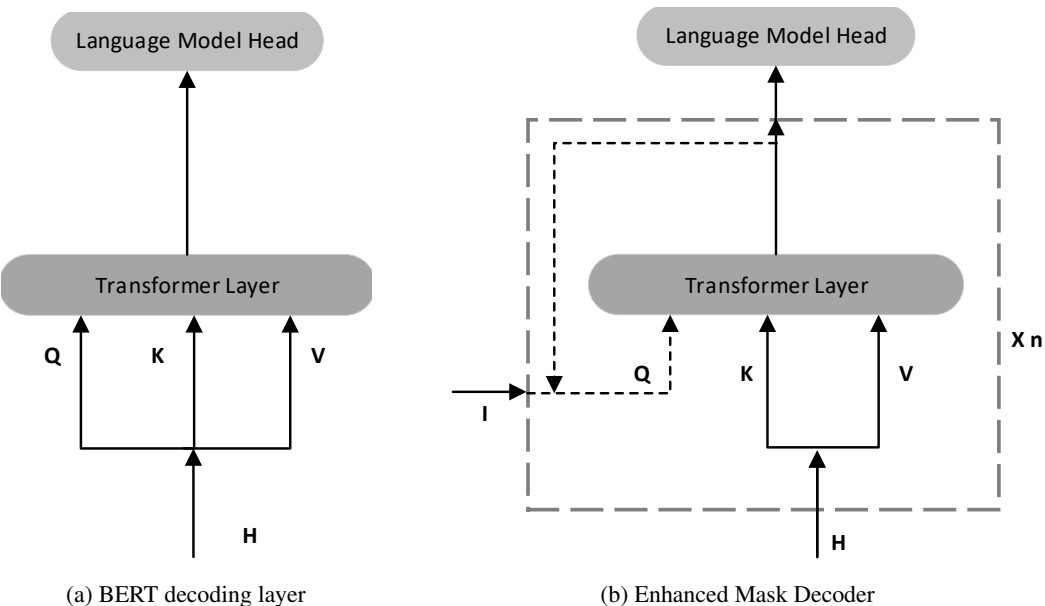

Figure 2: Comparison of the decoding layer.

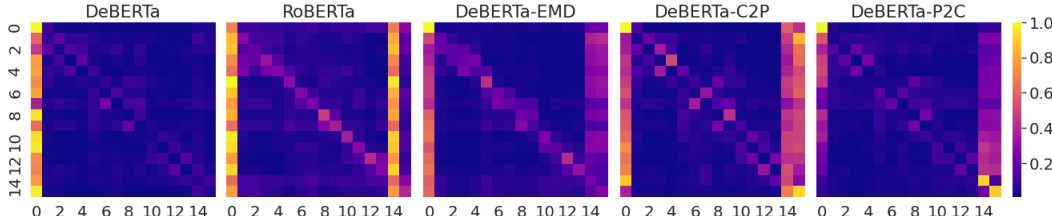

Figure 3: Comparison of attention patterns of the last layer among DeBERTa, RoBERTa and DeBERTa variants (i.e., DeBERTa without EMD, C2P and P2C respectively).

We observe two differences. First, RoBERTa has a clear diagonal line effect for a token attending to itself. But this effect is not very visible in DeBERTa. This can be attributed to the use of EMD, in which the absolute position embedding is added to the hidden state of content as the query vector, as verified by the attention pattern of DeBERTa-EMD where the diagonal line effect is more visible than that of the original DeBERTa. Second, we observe vertical strips in the attention patterns of RoBERTa, which are mainly caused by high-frequent functional words or tokens (e.g., "a", "the", and punctuation). For DeBERTa, the strip only appears in the first column, which represents the [CLS] token. We conjecture that a dominant emphasis on [CLS] is desirable since the feature vector of [CLS] is often used as a contextual representation of the entire input sequence in downstream tasks. We also observe that the vertical strip effect is quite obvious in the patterns of the three DeBERTa variants.

We present three additional examples to illustrate the different attention patterns of DeBERTa and RoBERTa in Figures 4 and 5.

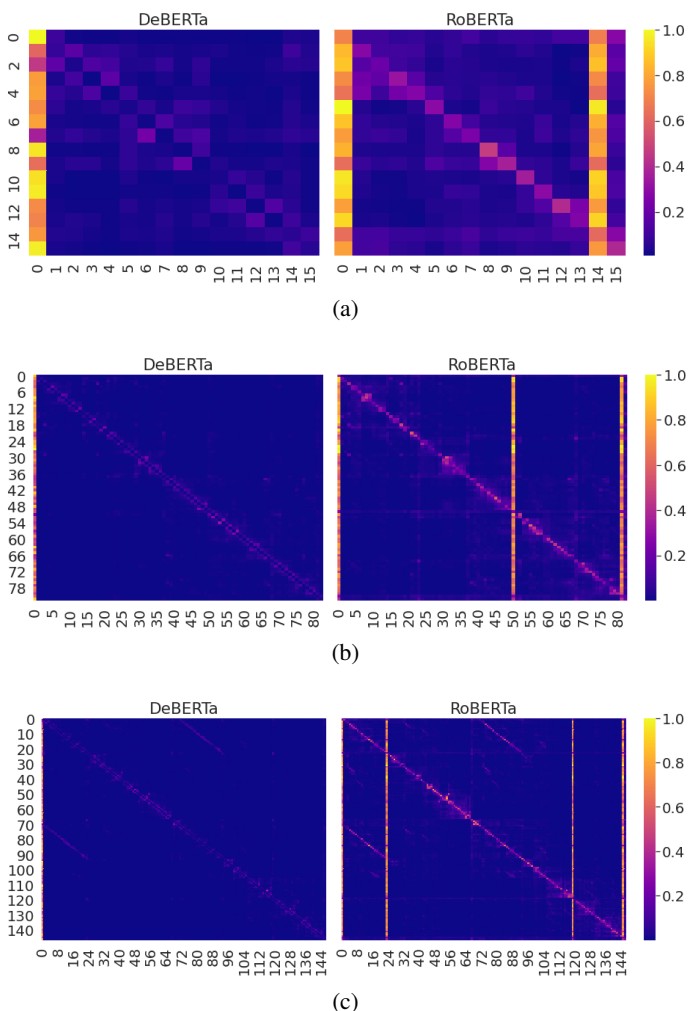

Figure 4: Comparison on attention patterns of the last layer between DeBERTa and RoBERTa.

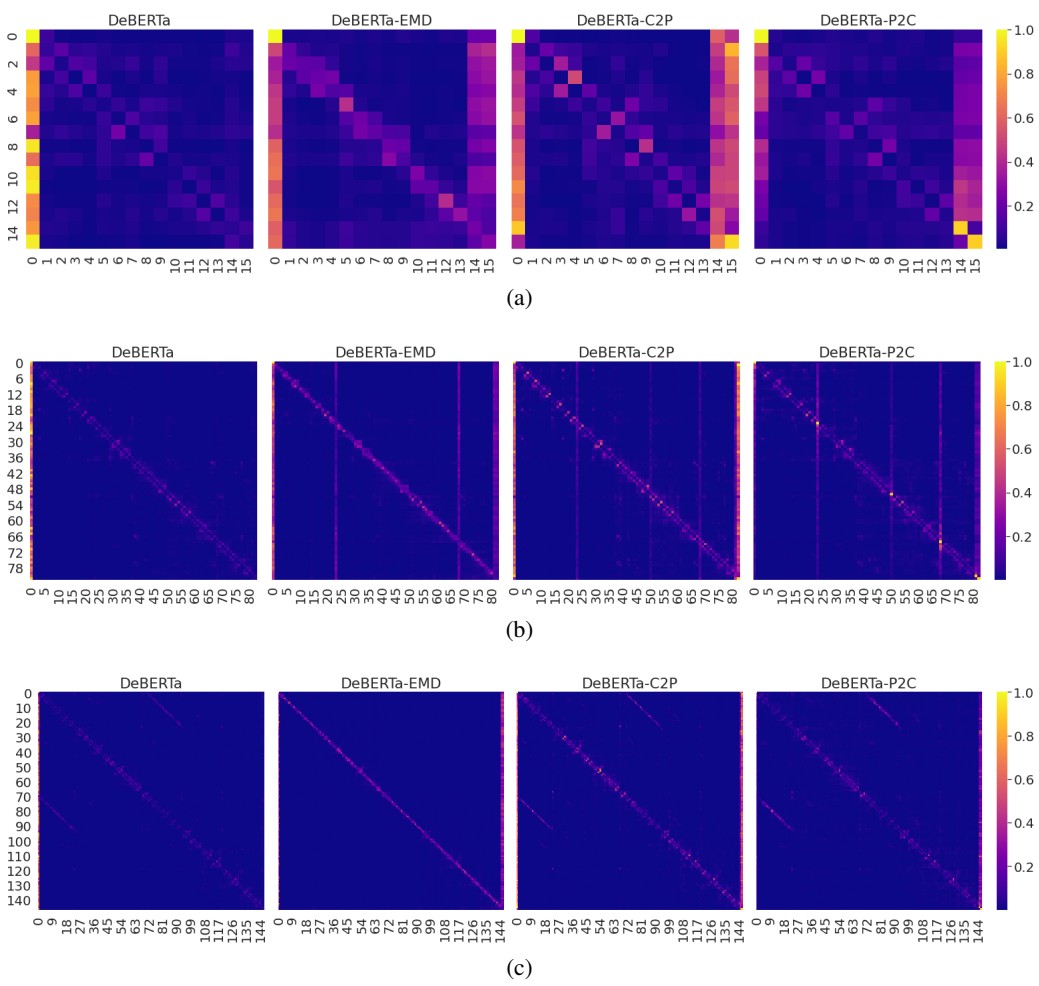

Figure 5: Comparison on attention patterns of last layer between DeBERTa and its variants (i.e. DeBERTa without EMD, C2P and P2C respectively).

## A.10 ACCOUNT FOR THE VARIANCE IN FINE-TUNING

Accounting for the variance of different runs of fine-tuning, in our experiments, we always follow Liu et al. (2019c) to report the results on downstream tasks by averaging over five runs with different random initialization seeds, and perform significance test when comparing results. As the examples shown in Table 14, DeBERTa$_{base}$ significantly outperforms RoBERTa$_{base}$ ($p$-value < 0.05).

| Model | MNLI-matched (Min/Max/Avg) | SQuAD v1.1 (Min/Max/Avg) | $p$-value |
|---|---|---|---|
| RoBERTa$_{base}$ | 84.7/85.0/84.9 | 90.8/91.3/91.1 | 0.02 |
| DeBERTa$_{base}$ | 86.1/86.5/86.3 | 91.8/92.2/92.1 | 0.01 |

Table 14: Comparison of DeBERTa and RoBERTa on MNLI-matched and SQuAD v1.1.

