# OpenReview forum: "DEBERTA: DECODING-ENHANCED BERT WITH DISENTANGLED ATTENTION"
_ICLR.cc/2021/Conference — ICLR 2021 Poster_

### Official Review · AnonReviewer2 · 2020-10-27
**DEBERTA: DECODING-ENHANCED BERT WITH DISENTANGLED ATTENTION**

**Rating:** 6
**Confidence:** 5

**Review:**

The paper proposes a BERT-inspired model that adds a two main different architectural decisions: different content and position representations (instead of a sum), and absolute positions in the decoding layer. The authors run the standard suite of GLUE benchmark experiments, on both “large” and “base” setups, as well as a generation setup (Wikitext-103).

The modifications proposed are not game-changing, but the evaluations are interesting in terms of understanding the impact of these modifications. One thing that I find disingenuous is fact that their disentangled approach does introduce additional parameters, which is not quantified (or even mentioned) in the main paper. I had to dig into the Appendix to see that this introduces about 49M additional parameters (increment of 13%).

Another problem that I have is with their experimental comparisons, especially the ones in main part, Sec 4.1.1. I’m listing below the most important issues in this section:

“RoBERTa and XLNet are trained for 500K steps with 8K samples in a step, which amounts to four billion passes over training samples”. This is confusing; what you mean to say is that the models see about four billion training examples. The term “passes” is used usually as an equivalent to “epochs”, ie how many times the model goes over the entire training set.

“[...] Table 1, which compares DeBERTa with previous models with around 350M parameters: BERT, RoBERTa, XLNet, ALBERT and ELECTRA.” Note that ALBERT is actually around 235M parameters, significantly less than all the others. You cannot simply bundle all together and claim they are equivalent parameter-size--wise.

“DeBERTa still outperforms them [ALBERT_xxlarge] in term  of the average “GLUE” score.” Note that the difference here wrt ALBERT_xxlarge is from 89.96 to 90.00, ie 0.04 for the average, with a tie 3-3 in terms of wins for specific tasks. Unless you can show that the 0.04 difference is statistically significant, you need to tone down the claim about “outperforming”.


“We summarize the results in Table 2. Compared to the previous SOTA models with similar sizes, including BERT, RoBERTa, XLNet and Megatron336M, DeBERTa consistently outperforms them in all the 7 tasks. Taking RACE as an example, DeBERTa is significantly better than previous SOTA XLNet with an improvement of 1.4% (86.8% vs. 85.4%).”
For whatever reason, the authors omit ALBERT from the comparison done for Table 2, in spite of its even smaller size compared to the included ones, and the fact that the ALBERT numbers for these tasks are readily available in the paper. Taking RACE as an example: ALBERT (single model) has 86.5% accuracy, therefore nullifying the claim of 1.4% improvement.


Re: References

A lot of the references use the Arxiv version for papers that have been peer-reviewed and published. Please fix.

---

> ### Author Response · Authors · 2020-11-24
> **Author Response to Reviewer 4**
>
> We would like to thank reviewer 4 for the detailed comments. Below we try to respond to the feedbacks mentioned in the review:
>
> **About additional parameters**. We will clarify this in the main paper. The additional parameters in our original model are due to the projection matrix of relative position embedding. We also perform a new model design via sharing the parameters of the two projection matrices, which makes the number of model parameters close to BERT or RoBERTa, without sacrificing the accuracy.  We report the experimental results in Table 11 in the Appendix.
>
> **About the experiment result description**. These are great feedbacks and we will describe the comparison more precisely in the new version, especially in the comparison with ALBERT. First, we will add the ALBERT result into Table.2.  Table 1 will focus on comparing models similar to BERT-large structure, i.e., 24 layers with 1024 dimensions and compare more SOTA models in Table.2, including ALBERT-XXLarge. Second, we will clarify the excellent design of the parameter sharing introduced by ALBERT, which can significantly reduce the model size although the computation cost is still determined by the model structure, i.e. number of hidden dimensions and transformer layers. As is reported in the ALBERT paper, the data-throughput of BERT-Large is about 3.17x higher compared to ALBERT-XXLarge.  We agree ALBERT-XXLargeand DeBERTa-large are comparable in terms of accuracy, with ALBERT-XXLarge having less model parameters  and DeBERTa being trained more efficiently as shown in Figure 1.
>
> Meanwhile, we will make the change to 4B training samples and fix the references with their latest updates.

---

### Official Review · AnonReviewer4 · 2020-10-28
**In this paper, an improvement of BERT model is proposed. It relies on the disentanglement of contents and relative positions in the encoding layers and on the incorporation of absolute positions in the decoding layer.**

**Rating:** 7
**Confidence:** 4

**Review:**

In this paper, an improvement of BERT model is proposed. It relies on the disentanglement of contents and relative positions in the encoding layers and on the incorporation of absolute positions in the decoding layer.


Strengths:
* The paper is well written, the positioning to the state of the art is clear and the method is rigorously described.
* The paper provides a complete evaluation using the existing benchmarks for NLP and including ablation studies and evaluation of pre-training efficiency and Deberta improves results in the major part of the cases.


Weaknesses:
* The proposed method is a relative increment of previous methods.
* In Section 4.1.1., the way performance increase or decrease is reported is not exact (1.1% -> 1.1 points)
* Do we have an idea of the statistical significance of the improvements?
* It would be interesting to have the rationale for the mitigated result obtained on Table 1. Is Deberta more relevant for specific tasks?
* The authors claim that they evaluate their results on generation task but it rather seems that they evaluate language modeling using perplexity.
*The use of non documented acronyms (ppl, for example) that could be not understandable outside the NLP community.
*They are some redundancy in the text (second paragraph of 3.2 and fourth paragraph of the introduction) that is not necessary.

---

> ### Author Response · Authors · 2020-11-24
> **Author Response to Reviewer 3**
>
> We appreciate the review and thank reviewer 3 for the thoughtful feedback.
>
> **About the incremental of previous methods**. We agree that our approach is an extension to previous methods. Besides a more comprehensive way to capture both the content and position, the main contribution of this paper is a detailed empirical study to demonstrate that the two proposed techniques (Disentangled attention and EMD) are simple and effective.
>
> **About statistical significance of the improvements**.  We perform a t-test on the MNLI and SQuAD V1.1 between DeBERTa and RoBERTa on their base models. The p-value on both datasets is less than 0.05.  More details are provided in one of the responses to the reviewer 2 above.
>
> **About the perplexity**. We follow previous work such as XLNet to report the perplexity.  But we will add a new generation task of next-word-prediction in the new version, as a complement to perplexity in the generation tasks.
>
> Meanwhile, we will incorporate other great feedbacks and fix them in our next version, including the notation in section 4.1.1, acronyms, and some redundancy in text.

---

### Official Review · AnonReviewer3 · 2020-10-28
**DeBERTa: Decoding-Enhanced BERT with Disentangle Attention**

**Rating:** 6
**Confidence:** 3

**Review:**

Summary and Contributions

The authors proposed an extension to the word representation transformer architecture  that takes into account disentangle features for position and content. The disentangle of attention is based on the composition of a content and position parameter matrices, in addition with combinations of both.  The main contribution is to tackle issues with the relative position embeddings used on standard transformer architectures. The proposed model shows improvements on some benchmarks by using less pre-training data compared to the baseline.

Strengths

- The proposed model tackles a known issue in transformer architectures.
- The authors perform a comprehensive comparison on standard text benchmarks as well as an ablation study.
- The findings show that disentangle attention improves results on some text benchmarks.

Weaknesses

- Related work on disentangle representations for text, and the further motivation for using disentanglement into the attention model are not discussed.
- Missing results of the variance in metrics with multiple runs on the downstream tasks. As an extra contribution, the authors could  show if the improvements are due to the proposed model or variance in parameter initialisation.

Questions to the Authors

- Could you elaborate on disentangled representations and how they relate to the proposed attention model?
- How does it compare the enhanced masked language model with the masked language model?
- How does the relative position parameter matrix is initialised, and how does it affect the language model performance?

---

> ### Author Response · Authors · 2020-11-24
> **Author Response to Reviewer 2**
>
> Thank you for the positive review. We provide the answer to the questions and potential concerns.
>
> **Q1**: The disentangled attention in DeBERTa is motivated but not closely related to disentangled representations or features.  Unlike the conventional absolute position bias encoding which adds the position embedding into content embedding directly, we borrow the idea of disentangled representations to decompose the attention score into different parts to avoid the interference between content and position, as well as fully capture the interaction between content and relative positions, and add the absolute position embedding back into the EMD layer in DeBERTa. We will add a footnote in new version to clarify this.
>
> **Q2**: About the difference between EMD and masked language model (MLM), we add absolute position encoding at the last layer in EMD to address the limitation of relative position encoding on MLM, which is demonstrated by the example in section 3.2 with an ablation study in Table.5.
>
> **Q3**: Following previous works such as BERT and RoBERTa, in our experiments, we simply use the conventional way to initialize our model (parameter matrices) using normal distribution N(0, 0.02).  How the initialization affects the model performance is an open research topic beyond the scope of this paper. We agree that it could be an important research topic for the future work.
>
> **About the variance with multiple runs**. Following BERT and RoBERTa, our reported numbers are based the average on 5 runs with different random initialization seed. Here are the results with min, max, average, and a t-test on MNLI, SQUAD for base model, as a complement to the Table.5 in the paper.
>
> ---------------------------------------------------------------------------------------------------------------------------------------
>
> |	                                                       |DeBERTa base(Min/max/avg)	| RoBERTa-ReImp base(Min/max/avg) |p-value of t-tests |
> |:---------------------------------------------:|:-------------------------------------------------:|:-------------------------:|:------------------------------:|
> | MNLI-matched(Acc)	                       |86.1/86.5/86.3                                         |	84.7/85.0/84.9	| 0.02 |
> | SQUADv1.1(F1)	                               | 91.8/92.2/92.1	| 90.8/91.3/91.1|	0.01 |
> ---------------------------------------------------------------------------------------------------------------------------------------
>
> In our paper all the improvements that we claimed statistically significant are based on statistically significant tests with p-value < 0.05.

---

### Official Review · AnonReviewer1 · 2020-10-30
**Good empirical performance but requires a more careful comparisons to prior works.**

**Rating:** 6
**Confidence:** 3

**Review:**

The paper proposed a novel attention mechanism and a new objective function that mitigates the distribution shifts caused by masked tokens for downstream tasks in MLM. It demonstrates superior performance across benchmarks.

Pros:
1. Good empirical results are demonstrated across an extensive suite of benchmarks. Ablation studies are well done. Hence I am willing to give a score of 6 despite of the following concerns.

Cons:
1. My major concern is about the novelty of this paper.
In transformer-XL[1], the idea of relative positional information in the form of Eq (2) was already introduced. The paper somehow intentionally omit the discussion following (2), only mentioning two earlier works of (Shaw et al., 2018; Huang et al., 2018). I think the author should be honest and compare with relative positional information introduced in transformer-XL in the forefront.
That being said, there is obviously still differences between transformer-XL and the proposed methods. And also the introduction of novel objectives in addition to the attention mechanism.

2. However, the previous concern brought up the second concern I have about the evaluations. Since the modification relative positional information of transformer-XL to the proposed method is not too large, I wonder if there is a reason to explain the better performances of the proposed methods. Hence I am worried if the baseline such as XLNet was well-tuned. We can see that for example in [2], the performance of XLNet was much better than originally reported. I think the author should try to carefully evaluate the relative positional mechanisms of prior works with authors' own infrastructure, while having everything else fixed.

3. I find the word "disentangled" a bit misleading in this context. Disentanglement in ML [3] often refers to the ability to disentangle factors of variations of the data. The work does not make use of any disentangled techniques, or have disentanglement representation/architectures. It simply use a relative position mechanism that's the sum of four matrix products.

[1] Dai et. al. Transformer-XL: Attentive Language Models Beyond a Fixed-Length Context

[2] Dai et. al. Funnel-Transformer: Filtering out Sequential Redundancy for Efficient Language Processing

[3] Locatello et. al., Challenging Common Assumptions in the Unsupervised Learning of Disentangled Representations

---

> ### Author Response · Authors · 2020-11-24
> **Author Response to Reviewer 1**
>
> We would like to thank reviewer 1 for the thoughtful comments and suggestions. Below we address the concerns mentioned in the review:
>
> **Difference with Transformer-XL**. We will clarify the commonness and difference in the next version. The relative position in DeBERTa is an extension of that in Transformer-XL and XLNet, with a different motivation and implementation. First, in Transformer-XL/XLNet,  the relative position is introduced to solve the position dependencies between tokens among different segments.  In DeBERTa,  the motivation is to decompose position information from content information thoroughly. Second, DeBERTa separates the content and position in a more comprehensive way.  For example, DeBERTa contains a new position-to-content component that captures the relative interaction between position and content at the attention layer. This is an important introduction in DeBERTa. As we showed in Table.5, this new component is critical in the new disentangled attention and can substantially boost the model performance in the Ablation study. Meanwhile, we did compare with XLNet in Table.5 and briefly mentioned the DeBERTa minus P2C will be reduced to XLNet plus EMD.  We will make this clearer in the revision.
>
> **The word Disentangled**. Thanks for the suggestion.  We will add a footnote in the paper to distinguish these two concepts.  Our disentangled attention is motivated but not closely related to disentangled representations or features.  Unlike the conventional absolute position bias encoding which adds the position embedding into content embedding directly, we borrow the idea of disentangled representations to decompose the attention score into different parts to avoid the interference between content and position, while fully capturing the interaction between content and relative positions. We will make this clear in our new version.

---

### Comment · ~Jonathan_Pilault1 · 2021-01-18
**SiFT (virtual adversarial training) mentioned in your arXiv version but not in the ICLR article: Was it used?**

Very interesting work.

In your arXiv version [1], SiFT (perturbations to the normalized word embeddings) was used. You wrote that "we find that the normalization substantially improves the performance of the fine-tuned models". Since the results of your arXiv version and the ICLR openreview version are the same for $DeBERTa_{large}$, I was wondering if you had used SiFT here. If so, could you discuss the performance differences between $DeBERTa_{large}$ with and without SiFT.

Thank you!

[1] https://arxiv.org/pdf/2006.03654.pdf

---

> ### Comment · ~Pengcheng_He2 · 2021-02-05
> **RE: SiFT (virtual adversarial training) mentioned in your arXiv version but not in the ICLR article: Was it used?**
>
> Thanks for your interest in our work. SiFT is only used with 1.5B model in SuperGLUE tasks in the paper. We will clarify this in our updated version.

---

### Decision · Program_Chairs · 2021-01-07
**Final Decision**

**Decision:**

Accept (Poster)

**Comment:**

All reviewers gave, though not very strong,  positive scores for this work.  Although the technical contribution of the paper is somewhat incremental, the reviewers agree that it solidly addresses the known important issues in BERT, and the experiments are extensive enough to demonstrate the empirical effectiveness of the method.  The main concerns raised by the reviewers are regarding the novelty and the discussion with respect to related work as well as some unclear writings in the detail,  but I think the pros outweigh the cons and thus would like to recommend acceptance of the paper.

We do encourage authors to properly take in the reviewers' comments to further polish the paper in the final version.